# Effects of Early Neglect Experience on Recognition and Processing of Facial Expressions: A Systematic Review

**DOI:** 10.3390/brainsci8010010

**Published:** 2018-01-06

**Authors:** Victoria Doretto, Sandra Scivoletto

**Affiliations:** Departamento de Psiquiatria, Hospital das Clinicas HCFMUSP, Faculdade de Medicina, Universidade de Sao Paulo, 05403-010 Sao Paulo, SP, Brazil; sandra.scivoletto@fm.usp.br

**Keywords:** child neglect, facial recognition, facial emotion, child maltreatment, review

## Abstract

**Background:** Child neglect is highly prevalent and associated with a series of biological and social consequences. Early neglect may alter the recognition of emotional faces, but its precise impact remains unclear. We aim to review and analyze data from recent literature about recognition and processing of facial expressions in individuals with history of childhood neglect. **Methods:** We conducted a systematic review using PubMed, PsycINFO, ScIELO and EMBASE databases in the search of studies for the past 10 years. **Results:** In total, 14 studies were selected and critically reviewed. A heterogeneity was detected across methods and sample frames. Results were mixed across studies. Different forms of alterations to perception of facial expressions were found across 12 studies. There was alteration to the recognition and processing of both positive and negative emotions, but for emotional face processing there was predominance in alteration toward negative emotions. **Conclusions:** This is the first review to examine specifically the effects of early neglect experience as a prevalent condition of child maltreatment. The results of this review are inconclusive due to methodological diversity, implement of distinct instruments and differences in the composition of the samples. Despite these limitations, some studies support our hypothesis that individuals with history of early negligence may present alteration to the ability to perceive face expressions of emotions. The article brings relevant information that can help in the development of more effective therapeutic strategies to reduce the impact of neglect on the cognitive and emotional development of the child.

## 1. Introduction

Man is naturally a social animal (Aristotle, 350 B.C. E.; Spinoza, 1677). Social interaction constitutes an important condition for a good quality of life, as well as for surviving and adapting in the world [1,2,3,4]. Faces provide a broad range of information and the ability to process such stimuli is critical for establishing successful relationships [5]. The information extracted while perceiving a face can be grouped into two categories: invariant and changeable aspects of face. Invariant aspects include visual features of the face that have relative stability over time, such as gender and identity. Changeable aspects of face include dynamic and transient facial cues, such as emotional expressions and intentions [6]. The perception of facial expression represents the recognition of distinct expressions, and understanding the meaning of facial expressions. Being able to precisely identify an individual identity is important for social relationships, once we interact with a large number of different people. However, the perception of facial expressions of emotions exerts a much more relevant and refined function in social communication [6]. 

### 1.1. Experience Drives Cortical Specialization

Most researchers agree that human face perception abilities reflect in part an innate, specialized face processing mechanism, but mostly a phenomenon of visual learning [7,8]. Development of the specialization for faces occurs through extensive experience and demands adequate exposure to facial expressions [9,10,11]. It begins early in life and continues through adolescence [12], in a process in which the neural circuitry becomes increasingly refined and develops towards a more mature form. Haxby and colleagues suggested a model of an organized neural system that involves the perception of facial traits. The model proposes that face perception is mediated by a neural system comprised of a set of regions in both brain hemispheres. The system involved in coding perception of emotional expressions consists of the inferior occipital region, the superior temporal sulcus (STS) and additional regions associated with emotion, including limbic regions, and insula and inferior frontal cortex [6,13].

### 1.2. Deprivation Alters the Neural System Involved in Recognition of Facial Expressions

Child neglect occurs when the person who is responsible for the child fails to provide the required physical, emotional and educational care, such as shelter, food, medical care and affection [14]. The child under this condition may encounter emotional and psychosocial deprivation, and a lack of sensory and cognitive stimulation [15,16]. Child neglect is the most chronic and prevalent form of child abuse [17]. It is often difficult to recognize since it frequently involves an absence of behavior, instead of the presence [14,18,19].

The initial impact of neglect may not be obvious, but the consequences of child neglect may be serious. Research shows child neglect can have a negative impact on the development of the following areas: emotional and psychological [20,21,22,23,24,25,26]; social and behavioral [27,28,29]; and health and physical [30]. There is a growing body of evidence indicating the potential impact of childhood neglect on structures and function of the brain. A recent review suggested that a lack of exposure to psychosocial experiences would lead to a reduction of dendritic arborization and density during neurodevelopment. It would result in a reduction of volume and cortical thickness of certain brain regions, as well as decreased performance on these areas [31]. It is also hypothesized that abusive experiences would release stress-induced hormones and neurotransmitters, resulting in impairment of neurogenesis, overproduction of synapses and pruning during sensitive periods in the developing brain [32,33]. These effects would likely target specific stress-susceptible brain regions in genetically susceptible individuals [33]. There is relatively consistent evidence for reduction in cortical grey and white matter volume [34], reduced orbitofrontal cortex [35] and increased amygdala volume in individuals who have experienced deprivation [36]. In terms of functional findings, recent literature suggest that experience of deprivation is related to hypoactivity in a number of brain regions, such as areas of the pre-frontal and temporal cortex, and limbic and paralimbic systems [37]. Many of these regions are involved in emotional regulation and facial emotion processing [33,38]. It is also known that the impact of neglect is variable according to the presence of protective factors, home environment, poverty [39], timing and duration [40]. 

### 1.3. Development of the Ability of Recognition of Emotional Expressions

A range of aspects of our perceptual and emotional abilities are heavily shaped by experiences [41]. Sensitive periods represent a limited period in development whereas the influence of experience on brain circuits is especially strong. Some capabilities, such as binocular vision, can only be developed within critical developmental periods, a specific type of sensitive period whereas the developing brain can be profoundly modified in a permanent way by experiences [41,42]. Such critical periods were described for some aspects of face perception. Children born with bilateral congenital cataracts, with subsequent deprivation of visual information for the first weeks of life, showed long-term impairment in their ability to recognize individuals’ identities, even after many years of regular exposure to human faces [43]. At present, there is no evidence of such critical periods for the recognition of emotional expressions in humans. 

Until recently, the impact of neglect on facial emotion recognition and processing has not been well summarized. Earlier studies on this subject tended to focus specifically on individuals with histories of physical or sexual abuse or multiple types of maltreatment [44,45]. General child maltreatment and physical abuse were associated with some form of alteration to face recognition and processing of emotions [45,46]. However, few studies have limited recruitment to individuals exposed to only neglect [8,47].

Better understanding of the impact of negligence on emotion recognition is fundamental for the development of more effective strategies for the treatment and rehabilitation of this population. This study provides a systematic literature review of recent research of child neglect and its impact on facial emotion recognition and processing. Our first hypothesis is that individuals with history of early negligence may present alteration to the ability to recognize and process face expressions of emotions. Possibly the alteration would be a consequence of the impact of deprivation on the brain system involved in perception of facial emotions. We also hypothesize that there is a sensitive period during which exposure to neglect impacts recognition and processing of faces in a more significant way. Finally, we hypothesize that distinct types of neglect experience have an impact on face recognition and processing in different ways, as a result of cortical specialization. 

## 2. Materials and Methods

### 2.1. Literature Search

A systematic search was conducted in the following databases in May 2016: PubMed, PsycINFO, ScIELO and EMBASE. We used the following search terms: “emotional recognition” OR “facial expression” AND “child neglect” OR “child, institutionalized”. To limit our search to the most recent studies, the search filter was set to select publication in the last 10 years. The searches were limited to English language. All abstracts were reviewed by the two authors independently and selected if they satisfied the criteria of being original studies of emotional face processing or emotional face recognition in children, adolescents or adults with history of childhood neglect. Those articles that did not present enough information to determine eligibility were selected for further review. Perception of facial emotions represents the recognition of face expressions and is mediated by a distributed neural system that processes such information. We considered studies of emotional face recognition that assessed accuracy for identification of facial expressions of emotions. For emotional face processing we considered those studies that evaluated neural mechanisms or attentional bias involved in the processing of face recognition.

### 2.2. Study Inclusion and Exclusion Criteria

After initial selection, the first author (V.F.D.) reviewed the full text of selected articles and selected the articles according to the following electing criteria: (a) original studies assessing samples of individuals with history of neglect (i.e., not reviews, meta-analyses, editorials, or case studies); (b) specific data of the impact of childhood neglect on face recognition and processing (or data of neglected groups separately of other types of child maltreatments or patient groups); (c) neglected children identified either through institutions or social services or by the use of a standardized assessment procedure for neglect characterization; (d) research that used instruments with presentation of face images in the assessment of emotional recognition (accuracy in identification of facial expression) and face processing (neural or attentional bias aspects involved in face recognition). Studies without available data to estimate the impact of childhood neglect were excluded. All included articles were reviewed by the other author (S.S.).

### 2.3. Data Analyses

Relevant data was extracted from studies and contained sample characteristics, such as age and size, instruments, emotion type and main findings. When available, data on accuracy of identification of facial expressions was abstracted.

Most studies clearly separate individuals with a history of negligence from those without neglect. However, the groups without history of neglect were quite heterogeneous among the studies: some were composed of healthy individuals, others by individuals with history of other types of maltreatment. Thus, in this review we will adopt the terms “neglect group” to designate individuals with a history of neglect and “comparison group” for individuals without negligence but who may have otherwise suffered other forms of abuse.

## 3. Results

### 3.1. Literature Search

A total of 1033 abstracts were identified. Of these, 45 studies were selected for more detailed evaluation. Figure 1 presents the summary of study selection process. Fourteen studies met inclusion criteria and their main characteristics and findings are described in Table 1. 

### 3.2. Samples Characteristics

General sample characteristics of the studies and main findings are described on Table 1. Data were available for approximately 1088 individuals with history of negligence and 1561 individuals of the comparison group. The sample sizes of the neglect groups ranged from 11 to 330, with a mean sample size of 77. Seven studies included participants younger than 18 years of age (mean age = 8.09 years) in their neglect groups and comparison groups (mean age = 7.89 years) and seven studies included adult participants in their samples (mean age = 39.5 years).

The characteristics of neglect experience varied substantially and included children who lived or were currently living in orphanages [48,49,50,51,52,53], children living on the streets [54], community children who experienced early neglect [55,56,57,58,59,60] and adults with self-report of early neglect. Most of the samples included healthy individuals, but some included individuals with co-morbid mental disorders, such as depression [56,57,60], anxiety [57] and bipolar disorder [58].

### 3.3. Studies’ Designs and Methodology of Assessment

One study assessed history of neglect by official records and six by standardized self-report procedures. The other seven studies considered the presence of neglect because of the context their sample had lived in: on the streets, in orphanages or in foster care. 

Most studies used cross-sectional emotion processing data, and one employed a longitudinal design to evaluate the persistence of the alteration over time [49]. 

Characteristics of the methods of assessment are described in detail in Table 1. In general, these studies evaluated positive, neutral and negative emotions. Three studies assessed six “basic” emotions—anger, fear, disgust, sadness, happiness and surprise [58,59,61] and one examined only one negative emotion valence [53]. Seven studies differentiated negative emotions in 3 valences: sadness, anger and fear. 

There were important variations in the testing paradigms and methods used to assess the recognition and processing of face expressions, including the type of stimuli used (intense or gradual images of facial expressions of emotions, static or dynamic images, individual or paired images), the length of stimulus presentation and the complexity of the tasks.

We identified three overarching themes for face perception of emotions in individuals with early neglect: (a) accuracy in emotional face recognition—methods that evaluated the accuracy of identification facial emotion expressions; (b) neural aspects in processing emotional faces—methods that evaluated the underlying brain system; and (c) attentional aspects in processing emotional faces. Six studies investigated the accuracy of facial emotion recognition by using facial expression recognition tasks [48,54,58,59,60,61]. Five studies employed specific methods to explore neurological aspects of perception of facial expressions [49,52,53,55,56] and three attentional aspects [50,51,57]. Tasks designed for identification of facial expressions involved random presentation of face images in a booklet or on a computer screen and required participants to label the emotional content of individual facial expressions. To examine the neural and attentional systems underlying face perception, studies sought to focus the participants’ attention on face, while other techniques were applied to access features of face processing. One functional neuroimaging study [53] and all attentional bias studies [50,51,57] requested the participant press a button whenever they identified a target appearing on the face/screen, while another study requested the participants identify the face´s gender [56], to passively look the at faces images [49,55] or to look the faces with a required attention condition [52]. While participants performed the task, four studies employed neuroimaging techniques such as functional magnetic resonance imaging (fMRI) [52,53,55,56] and one study used event-related potentials (ERPs)—a measure of the electrical activity of the brain manifested in response to minimal stimulus events [49].

### 3.4. General Findings

Twelve studies reported that early neglect is associated to some form of alteration to the perception of emotional expressions. Two studies didn’t find difference in the recognition of emotional faces in childhood neglect compared to controls [59,60]. Next, we are going to describe the main findings of the studies. 

#### 3.4.1. Accuracy in Emotional Face Recognition—Facial Expression Recognition

Six studies assessed accuracy of recognition of specific facial expressions in individuals with early neglect compared to those without neglect. There are mixed findings from these studies. 

Moulson and colleagues investigated Romanian children enrolled in the Bucharest Early Intervention Project (BEIP) at a mean age of 9.7 years. These institutionalized children have experienced an extreme form of psychosocial deprivation in institutions during early years of life. A task with facial stimuli that changed in intensity from neutral to extreme emotional expressions was used, making it highly sensitive to subtle deficits in emotion processing. They showed that institutionalized children were less sensitive in identifying happy faces [48]. A recent follow-up investigation of this study showed that when these children reached 12 years of age there was a maintained deficit in the processing of facial expressions. Institutionally reared children required more perceptual information to accurately identify happy facial emotions, when compared to family-reared children [62]. Ardizzi and colleagues assessed 5- to 10-years-old children living on the streets of Sierra Leone (mean = 7.65 years ± 1.68), frequently exposed to a very abusive and neglectful environment. They found that street children recognized significantly fewer facial expressions of fear and sadness, but recognized more angry facial expressions than controls [54]. Furthermore, another study assessed the abilities of 4-year-old community children to label and recognize emotions in a three-component task. Those with history of neglect, confirmed by Child Protective Services, showed poorer performance in the combined tasks [61].

Three studies evaluated accuracy in perceiving emotional faces in adult individuals. They assessed childhood neglect retrospectively by a self-administered instrument and information about age and duration of exposure to neglect were not available. Russo and colleagues found that adults with bipolar disorder and childhood neglect showed decreased accuracy in identifying anger compared to those without history of early neglect [58]. Germine and colleagues provided data of anonymous visitors to a website, in which the task was self-administered via the Internet [59]. They did not find alteration to the recognition of facial expressions in individuals with childhood neglect. Another study evaluated adults with and without depressive symptomatology and also found no difference between neglected individuals and comparison [60].

#### 3.4.2. Neural Aspects in Processing Emotional Faces—Emotional Face Processing

##### Functional Magnetic Resonance Imaging

We identified four functional magnetic resonance imaging (fMRI) studies assessing brain activity in response to face expressions. All reported that amygdala showed greater activity during the processing of some facial expressions, including fearful [53], angry [52] and sad faces [55,56]. Increased amygdala activation to fearful faces was observed in 9.3-year-old internationally adopted children. These children had a mean age of 2.8 months when placed in orphanage and 17.8 months when adopted by their families [53]. Elevated activation toward fearful and angry faces was also observed in 9- to 18-year-old adopted youths with previous history of foster care or institutionalization. They were 1 to 72 months old when they were first placed into foster care or international orphanages and had been living with in their adoptive families for a mean of 8 years [52]. Furthermore, findings from these studies showed different patterns of activation of other brain regions to fearful faces, including greater hippocampus activation [52] and relatively greater activation of ventro medial prefrontal cortex (vmPFC) [53]. We identified 2 studies reporting fMRI findings in adults with history of childhood neglect assessed by self-report. Dannlowski and colleagues showed heightened amygdala excitability in response to subliminal sad faces in adults free from any lifetime history of psychiatric disorders [55]. Grant and colleagues demonstrated a positive but weak correlation between neglect and increased right amygdala response to sad faces in a sample including healthy and depressed individuals [56]. 

##### Event Related Potentials

We identified only one study that used ERPs to examine brain activity in response to face expressions of happy, angry, fearful, and sad. They assessed 136 institutionalized children in Romanian orphanages and comparison between 5 and 31 months of age at three time points: baseline, 30 months of age and 42 months of age. Almost half of these children had lived all of their lives in the institutions. The study found that neural responses toward emotional faces were similar between institutionalized and family-reared children, thus neglected children did not present atypical neural processing of facial expressions. Although they detected that children who experienced institutional care had reduced ERP amplitudes and latencies for occipital components compared to never-institutionalized children that persisted until at least 42 months of age, it seems unlike that only emotional process deficits explain these findings once these children also present many other cognitive deficits. After placement in foster care, previously institutionalized children showed intermediate amplitudes and latencies for occipital components, between the institutionalized and never-institutionalized children, at the 30- and 42-month assessments. It is important to note that the improvement of cortical hypoarousal was not associated with the age at which children were allocated into foster homes [49].

#### 3.4.3. Attentional Aspects in Processing Emotional Faces

##### Visual Attention Bias

Three studies assessed visual attention bias toward emotional faces. The attentional bias was accessed through a dot-probe task, a paradigm in which children had to correctly indicate the location of a stimulus after the presentation of a pair of emotional faces. 

Two studies provide data of children abandoned in infancy and placed into Romanian institutions who enrolled in the Bucharest Early Intervention Project (BEIP). Age of placement into foster care home ranged from 7 months to 8 years of age. Troller-Renfree and colleagues (2015) reported a significant attention bias to angry faces in institutionalized children at 8 years, but those who were placed into foster care showed a positive bias to happy facial expressions. They also reported that younger ages of placement into foster care was related to increased positive attention biases [50]. Similarly, Troller-Renfree and colleagues (2016) [50] found that those children taken out of an institution and placed into foster care maintained a significant positive bias to happy facial expressions at 12 years of age. Greater time spent in institutional care was correlated to increased biases towards threat [51]. 

Another study investigated attentional bias to emotional faces in depressed adults with early adverse experiences. They found that physical neglect was related to attentional bias toward sad faces [57]. 

## 4. Discussion

To our knowledge, this is the first review to examine specifically the effects of early experience of neglect alone, without other forms of maltreatment, on the recognition and processing of face expressions. We have identified 14 recent studies assessing the impact of early neglect on recognition and processing of facial expression in children and adults with 12 indicating significant impact. However, the results were conflicting and we detected a significant heterogeneity of methods, sample frame, instruments and type of expression across the studies. 

### 4.1. General Findings

The results differed significantly across studies. The inconsistencies across these studies may result from implementation of distinct instruments and methodologies and from differences in the composition of the samples, such as size, type of neglect experience, age and period of exposure and presence of adversities other than neglect [32]. All these variables interfere with the accuracy of comparisons and conclusion of results, such as those from meta-analytical studies. 

### 4.2. Neglect and the Impact on Neural Systems Involved in Processing Facial Expressions

Most studies of this review reported some form of alteration to the neural and attentional system involved in processing face expression in individuals exposed to childhood neglect. The specific type of alteration differed significantly across studies. There was a consensus among the studies that used fMRI to access amygdala activity during exposure to face expressions showing heighted amygdala activation in response to negative emotions [52,53,55,56]. Finally, the three studies that accessed visual attention bias showed some type of attentional bias in those individuals with childhood neglect [50,51,57]. These findings support our hypothesis that experience of neglect alters the neural system involved in processing facial expressions with subsequent impact in the ability to correctly recognize face expressions. In contrast, electroencephalographic (EEG) measures of children who experienced early neglect in institutions showed a normative response pattern toward emotional faces. One possible explanation is that facial emotion processing may be relatively spared in institutionally reared children, once the exposure of type and quantity of facial expressions they receive may be enough to establish a regular facial emotion recognition neural network. Another possibility is that facial emotion processing may be affected by institutionalization, but the facial emotion processing task used in the study was not sensitive enough to detect any alteration. There were two studies accessing accuracy of face recognition that did not find any alteration [59,60].

The amygdala is long known to play a central role in the neural regulation of emotion [63,64]. Mehta and colleagues (2009) reported increased amygdala volume in adolescents who experienced early institutional deprivation [65]. Previous literature suggests that when the amygdala identifies an emotionally relevant stimuli, it transmits a signal to the cortical areas that leads to an increase of attentional state [66]. Greater amygdala activity has been associated with amplified vigilance toward emotionally significant stimuli [67]. One possible explanation is that the increased amygdala activity toward negative faces reported in neglected individuals may interfere with attentional bias and accuracy in recognizing face expressions. 

### 4.3. Critical Periods and Face Perception of Emotions

Whether there is a sensitive or critical period for perception of face expressions remains an open question. In this review, one longitudinal study suggested the presence of sensitive period for attention bias in neglected children, although it has not identified a delimited specific time period. In the mentioned study, institutionalized children were placed into foster care when they had 207 to 2915 days of age, and those placed at younger ages showed a larger positive attention bias at 8 years old. They then concluded that either there is no sensitive period or, if there is one, it remains open beyond the age at which these children received the intervention.

However, there is a number of studies in literature that report sensitive periods of brain structures involved in the neural system of perception of emotions, including the amygdala, prefrontal cortex and occipital cortex in individuals exposed to childhood neglect [33]. Reports from the BEIP study found that children from 6 to 31 months of age living in institutions, at the baseline assessment, had relatively higher levels of theta band relative to children who had never been institutionally reared [68]. Some of the children were placed into foster care at ages ranging between 7 and 31 months, and EEG was again collected when they were 42 months and then at 8 years. The 42-month assessment showed that those children placed into foster care at ages younger than 24 months showed increasing alpha power relative to older-placed infants [69]. At 8 years, children placed into foster care demonstrated normalization of alpha activity, showing a pattern comparable to never-institutionalized children. Once again the effect on EEG alpha activity was greater in those institutionalized children placed into foster care before 2 years of age [70]. Thus, the timing effect in those findings suggests the presence of a sensitive period for neural activity in the face of severe neglect, although it is not yet possible to delimit the specific sensitive period. 

### 4.4. Type of Neglect Experience and Cortical Specialization

Some of the studies support our hypothesis that different types of neglect experience may impact facial recognition and processing system in different ways, influencing the ease of recognition of specific face expressions. For example, reduced ability to identify happy faces, but not to fear, sad or angry expressions, has been reported in individuals exposed to institutional care [49]. In contrast, children living on the streets showed no change in their ability to identify happy faces, but an increased ability to identify angry faces and reduced ability to label fear and sadness [54].

One possible explanation for these findings is that it reflects very diverse experiences, such as an inappropriate frequency of exposure to the full range of facial expressions, leading to specific cortical specialization for faces. Children reared in large institutions often experienced severe psychosocial deprivation [71] and may have been exposed to happy face expressions in an inferior frequency than children reared in typical environments. On the other hand, the life conditions experienced by street children is characterized by intense caregiver and material deprivation and exposure to high levels of violence, intimidation and physical assaults [72]. It could explain why these children were less skilled in recognizing sadness and fearful expressions, but had a better performance in labeling anger [54]. Pollak and colleagues (2005) suggested that children exposed to abusive homes become hypervigilant to anger in their environment once they learn to associate anger with danger, and consequently they engage in a constant state of anticipatory monitoring of the environment [73]. Furthermore, a recent study showed that street children display an inferior facial mimicry in response to negative facial expressions compared to non-neglected children. This finding suggests that a biological reduction of automatically evoked mechanisms is related to the form of responding to negative facial expressions [74].

However, another explanation is that these differences across studies may reflect a range of possible confounding variables other than the characteristics of neglect experience. Children reared in neglect contexts are also susceptible to the experience of a number of additional stressors, such as exposure to extreme violence in the family or community, poverty, and parental psychiatric disorders [32]. Therefore, it is difficult to attribute the alterations to a specific adverse rearing experience, such as negligence.

### 4.5. Limitations

We recognize that this systematic literature review should be interpreted with caution due to limitations. An important aspect is that there was methodological diversity among studies, sample constitution (different experiences of neglect, age of onset, duration of exposure) and control constitutions—the control groups were composed either of individuals without any history of maltreatment or by individuals who have suffered other forms of abuse, some of them with mental disorders. These differences made it impossible to carry out a meta-analysis of the studies. 

It is possible that the non-significant results detected by some of the studies were related to characteristics of the sample, such as lower levels of negligence, or to a low sensitivity of the tasks employed, instead of a lack of impact on recognition and processing of face expressions. Not surprisingly, two of the studies that did not find impairment in global or specific emotion recognition were studies that employed self-administered instruments to identify history of neglect. Also, sample size might be an issue: small sample sizes may result in underpowered studies. Finally, another limitation is that mental disorders, onset age and duration of exposure to negligence were not measured and controlled consistently as possible mediator of alterations to recognition and processing of facial expressions. 

## 5. Conclusions

This review was inconclusive due to the previously mentioned limitations, the methodological diversity between studies and sample differences. Despite these limitations, some studies supported our hypothesis that individuals with history of early negligence may present alteration to the ability to identify and process face expressions of emotions. Most of the findings reported alterations to the recognition and processing of negative emotions. Regarding our second hypotheses, there was no conclusive answers regarding whether there is a sensitive period during which exposure to neglect impacts perception of faces in a more significant way. Finally, some studies corroborated our hypothesis that different types of neglect experience have an impact on face perception of emotions in different ways.

This review brings relevant information that can help in the development of more effective therapeutic strategies to reduce the impact of neglect on the cognitive and emotional development of children.

Additional studies should address the methodological shortcomings discussed above, applying standard instruments and procedures to control the variables that influence perception of facial expression. They will also need to include large samples, assess performance across all six “basic” emotion expressions, and identify the effect of protective factors such as preventive and rehabilitation programs. Further research with neglected children, utilizing longitudinal designs to address the issues of alteration to recognition and processing of faces over time and the existence of a sensitive period for development of this ability, are vital.

## Figures and Tables

**Figure 1 brainsci-08-00010-f001:**
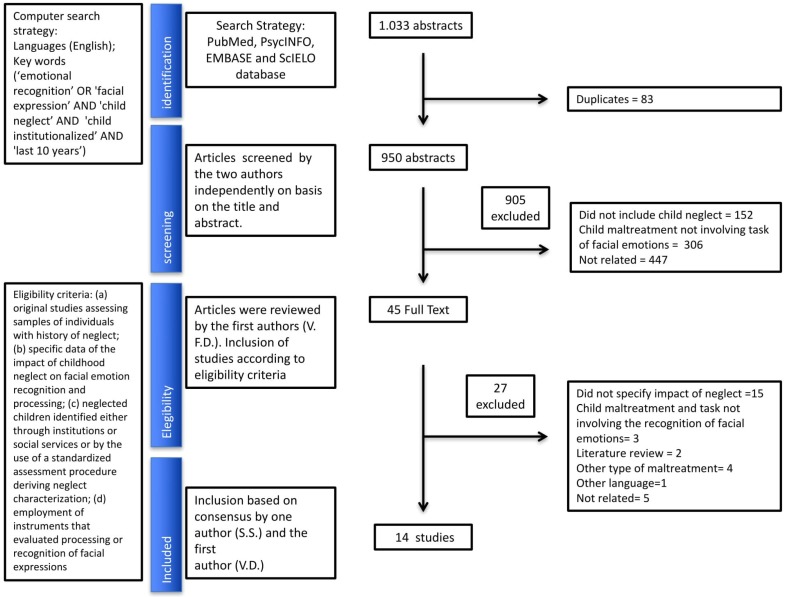
Flowchart of study selection process.

**Table 1 brainsci-08-00010-t001:** Characteristics and main findings of the studies of child negligence and its impact on emotional recognition and processing.

First Author	Neglected Sample (*N*)	Mean Age of Neglected Sample (years)	Control Sample (*N*)	Mean Age of Control Sample (years)	Neglect Accesses Through	Task	Type of Faces	Type of Stimuli	Data Acquisition	Main Finding
Dannlowski 2013 [55]	134	34.5	-	-	CTQ	View faces	happy sad neutral	Dynamic subliminal facial images on screen	fMRI	Increased amygdala activity toward sad faces
Grant 2011 [56]	10	39.3	26	30.25	CTQ-SF	Identify gender	happy sad neutral	Arrays of three images of emotions + target	fMRI	Increased amygdala activity toward sad faces
Maheu 2010 [52]	11	13.75	19	13.41	Experience in foster care or orphanages + K-SADS-PL	View faces	happy angry fearful neutral	Images of facial expressions + 4 attention condition	fMRI	Increased amygdala activity toward fearful and angry faces. Increased hippocampus activity toward fearful faces
Tottenham 2011 [53]	22	9.3	22	10.8	Experience in Institution	Identify target	fearful neutral	Images of facial expressions	fMRI	Increased amygdala activity toward angry faces. Increased vmPFC activity toward fearful faces
Moulson 2009 [49]	62	1.96	23	1.75	Experience in Institution	View faces	happy sad anger fearful	Images of facial expressions	ERP	Smaller amplitudes and longer latencies toward angry, fearful, sad and happy
Troller 2015 [50]	105	8.5	52	8.46	Experience in Institution	Identify target	happy sad neutral	Paired images of faces expressions + target	Attention Bias	IC displayed attention bias toward angry faces. FC displayed attention bias toward happy faces
Troller 2016 [51]	99	12.6	48	12.68	Experience in Institution	Identify target	happy sad neutral	Paired images of faces expressions + target	Attention Bias	FC displayed attention bias toward happy faces
Gunther 2015 [57]	17	34	-	-	CTQ	Identify target	happy sad	Paired images of faces expressions + target	Attention Bias	Attentional bias toward sad faces
Sullivan 2010 [61]	15	4.09	27	4.05	CPS records	Identify emotion	happy sad surprise mad scared disgust	Images of facial expressions	Accuracy	Impaired accuracy in emotion recognition represented by a single score in a combination for all emotions.
Ardizzi 2015 [54]	31	7.65	31	7.77	Street- children	Identify emotion	happy sad anger fearful	Dynamic facial images on screen	Accuracy	Decreased accuracy for fear and sad faces and increased accuracy for angry faces
Moulson 2015 [48]	65	9.7	67	9.78	Experience in Institution	Identify emotion	happy sad fearful angry neutral	Dynamic images of faces expressing gradual emotions	Accuracy	Higher detection threshold for happy faces
Russo 2015 [58]	36	48.2	39	46.2	CTQ	Identify emotion	happy sad angry disgust fear surprise	Dynamic images of facial expressions	Accuracy	Decreased accuracy for angry faces
Germine 2015 [59]	330	33.5	1170	33.5	Adapted ACES	Identify emotion	happy sad fearful angry disgusted surprised	Dynamic images of faces expressions	Accuracy	No difference in accuracy
Suzuki 2015 [60]	34	48.05	37	48.8	CTQ	Identify emotion	happy fearful sad angry	Dynamic facial images on screen	Accuracy	No difference in accuracy

IC: Institutionalized children; FC: Foster care children; CPS: Child Protective Services; CTQ: Child Trauma Questionnaire; CTQ-SF: Childhood Trauma Questionnaire– Short Form; ACES: Adverse Childhood Experiences Scale; K-SADS-PL: Schedule for Affective Disorders and Schizophrenia for School-Age Children—Present and Lifetime Version.

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
