# Peer review of "Effects of Early Neglect Experience on Recognition and Processing of Facial Expressions: A Systematic Review"

_brainsci, 2018, doi:10.3390/brainsci8010010_

Round 1

Reviewer 1 Report

The new version of the manuscript properly addresses most of the comments raised in the revision. I have only some minor points that I consider relevant before manuscript acceptance.

Line 22: Throughout the new version of the manuscript, the authors stressed the interest in neglect conditions without other forms of maltreatment. I totally disagree with this interpretation added after the review processing. It is extremely unlikely that neglect conditions occur alone without other forms of maltreatment. This assumption can also be confirmed considering the samples included in the revised studies. I ask authors to modify all these statements with a more realistic consideration of the actual maltreatment conditions (e.g., prevalent neglect condition).

Line 23: “due to methodological diversity”

Line 48-49: reformulate the sentence

Line 216 “Sierra Leonean

Line 218 [54]

Paragraph 4.4.: In my previous review I strongly asked authors to cite and discuss the results of Ardizzi, M., Umiltà, M. A., Evangelista, V., Di Liscia, A., Ravera, R., & Gallese, V. (2016). Less empathic and more reactive: the different impact of childhood maltreatment on facial mimicry and vagal regulation. PLoS one,11(9), e0163853. The authors completely misunderstand my point, they again cite the study already included in the revision (Ardizzi et al., 2015). In Ardizzi et al., 2016, a sample of street-children showed reduced facial mimicry in response to negative facial expressions of emotions. This aspect adds substantial information to the behavioral effects, which are the topic of the review. Indeed, it demonstrates that, at least among street-children, the explicit deficit in negative emotion recognition is associated with an implicit and physiological reduction of spontaneous mechanisms, like the facial mimicry. This result supports authors’ hypothesis about a different impact of the environment/experience between street-children and institutionalized children Please reformulate lines 354-357 as follow: “Pollak and colleagues (2005) suggested that in abusive home environments children learn to associate anger with threat of harm and therefore, they are hypervigilant to anger in their environment, maintaining a state of anticipatory monitoring of the environment as a possible mechanism of survival [78]. Coherently, street-children showed also a suppress spontaneous facial mimicry responses to negative facial expressions of emotions [Ardizzi et al., 2016 Less empathic and more reactive: the different impact of childhood maltreatment on facial mimicry and vagal regulation. PLoS one,11(9) doi: 10.1371/journal.pone.0163853. eCollection 2016.]

Please, update also the reference list adding Ardizzi et al., 2016 Less empathic and more reactive: the different impact of childhood maltreatment on facial mimicry and vagal regulation. PLoS one,11(9) doi: 10.1371/journal.pone.0163853. eCollection 2016.

Reviewer 2 Report

This systematic review examines the impact of childhood neglect on facial emotion processing. This addresses an important topic, and highlights remaining questions in the field regarding the extent to which facial emotion processing is affected by early neglecting experiences. I have a few suggestions for further improving the manuscript.

Section 1.2 is focused on how “deprivation alters the neural system involved in recognition of facial expressions”. However, the content in this section is more focused on the influence of stress. For example, on page line 67-68, authors discuss one theory for why neglecting experiences might alter the developing brain, in that increased release of stress-hormones interfere with neurogenesis, synaptic proliferation and pruning, and myelination. To focus this section in the influence of deprivation, authors may also emphasize here the factors beyond “stress exposure” that may explain disparities. Specifically, as authors discuss elsewhere in the manuscript, alterations in brain development may be due to a chronic absence of necessary input necessary for normative brain development to occur. This absence may occur during a sensitive period or chronically.

In section 1.3, authors discuss that there is limited evidence for the existence of critical periods in the development of emotion processing. Authors stated on line 89-90, “previous reports suggested the presence of sensitive periods, as it was detected longterm impact of neglect on recognition of faces expressions.” At least in terms of how this is described here, it is not clear to me that this definitively points to the potential existence of a critical or sensitive period. This previously documented long-term influence could just as well be due to chronic, long term exposure across multiple developmental periods, rather than due to missing input during a select period of development.    

When discussing the Moulson et al findings in section 3.4.1., I recommend that authors also cite the most recent follow up paper on this study: “Bick, J., Luyster, R., Fox, N., Zeneah, C., Nelson, C.A. (2016). Effects of early institutionalization on emotion processing in 12-year-old youth.” Developmental Psychopathology, 29, 1749-1761.”  This paper describes continuing trends of emotion processing deficits once the children reach 12 years of age. Also, in this section, or potentially elsewhere, authors may also want to discuss various ways of assessing facial emotion processing. In this study, we used a morphing task where a face was presented as absent to 100% intensity, and we assessed perceptual boundaries of emotion categorization (i.e. the point at which children were able to accurately discriminate the emotion from a series of morphed faces that varied in low emotion intensity to high emotional intensity).

In section 3.4.2: Regarding authors interpretation of the results from the ERP task: the study found that neural responses to emotional conditions did not differ between neglected and non neglected children, but that overall ERPs were reduced in magnitude in the neglected children, as authors mention, this was considered potentially due to delayed maturation, underarousal, lack of specialization of the cortex. Because there was no difference across emotion conditions, it is difficult to conclude that emotion processing deficits explain these findings. In fact, the BEIP interpreted these findings as evidence that facial emotion processing in institutionally reared children may be relatively spared, when compared to other areas of deficits observed in these children (attention, executive functioning). See for example, Moulson et al., 2015, cited in this paper. Authors may want to emphasize this conclusion in their discussion as well.

Also in the discussion, authors may want to talk about the fact that most cases of child abuse, at least in family settings or in “street children”, likely contain a heterogeneity of stressors- neglect, violence exposure, abusive caregiving, potential poverty, etc. Therefore, considering the multiple domains of adverse exposure may help to clarify discrepancies in the literature.
